# Prevalence of delayed treatment for sexually transmitted infections and its determinants in sub-Saharan Africa. A systematic review and meta-analysis

Muluken Chanie Agimas◉*, Milkias Solomon, Daniel Alayu Shewaye, Dessie Abebaw Angaw◉, Nebiyu Mekonnen Derseh

Department of Epidemiology and Biostatistics, Institute of Public Health, College of Medicine and Health Science, University of Gondar, Gondar, Ethiopia

* mulukensrc12@gmail.com

## Abstract

### Background

Sexually transmitted infection is a common public health issue, and it is characteristically transmitted through sexual intercourse. Around the globe, particularly in less developed countries, delayed treatment of this infection could lead to a health and economic burden. Even though the health and economic burden of sexually transmitted infections is high, studies to identify the pooled proportion and the possible factor of delayed treatment seeking are rare in sub-Saharan African countries.

### Objective

To assess the prevalence of delayed treatment for STIs and its determinants in sub-Saharan African countries.

### Method

Articles searched on search engines like Medline via PubMed, HINARI, Embase, Scopus, Cochrane Library, Science Direct, and websites like Google Scholar. The searching mechanism was using keywords and medical subject heading terms by combining the key terms of the title. To assure the quality of the included articles, Joana Brigg's Institute critical appraisal checklist was used. To assess the heterogeneity of the studies, a sensitivity analysis was conducted. The PRISMA checklist was used, and to estimate the pooled odds ratio, a random effect model was considered. The pooled odds ratio of 95% CL was used to identify the factors.

### Results

About 13 studies with 46,722 participants were incorporated. Despite considerable heterogeneity, the pooled prevalence of delayed treatment for STI in Sub-Saharan Africa was 47% (95% CI: 42%–51%, $I^2$ = 98.42, p<0.001). Geographically, the higher pooled prevalence of

**Data Availability Statement:** All relevant data are within the manuscript and its Supporting Information files.

**Funding:** The author(s) received no specific funding for this work.

**Competing interests:** The authors have declared that no competing interests exist.

**Abbreviations:** CL, Confidence Level; CI, Confidence Interval; OR, Odds Ratio; PRISMA, Preferred Reporting Item for Systematic Review and Meta-Analysis; STD, Sexually Transmitted Disease; STI, Sexually Transmitted Infection; WHO, World Health Organization.

delayed treatment for STI was in the eastern part of Africa (50%) (95% CI: 41%–59%, $I^2$ = 98.42, p<0.001). Rural residence (OR = 1.44, 95% CI: 1.03–2.01, $I^2$ = 39.3%, p-value = 0.19), poor knowledge about STI (OR = 1.49, 95% CI: 1.04–2.13, $I^2$ = 93.1%, p-value = <0.001), perceived as STIs not serious (OR = 2.1, 95% CI: 1.86–2.36, $I^2$ = 73.7%, p-value = 0.022), misconception for STD cause (OR = 1.39, 95% CI: 1.12–1.72), no education (OR = 4.1, 95% CI: 3.4–5.1), primary education (OR = 3.17, 95% CI: 2.23–4.2), and secondary education (OR = 1.57, 95% CI: 1.1–2.76) as compared to secondary and above education were factors associated with delayed treatment for STIs.

## Conclusion

The pooled prevalence of delayed treatment for STIs in Sub-Saharan African countries was high. Poor knowledge, attitude, and educational status affect the treatment delay for STIs. Thus, improving knowledge, educational status, and attitude are highly recommended to reduce the delayed treatment of STIs.

## Introduction

Sexually transmitted infection is a common public health issue, and it is characteristically transmitted through sexual intercourse [1–4]. The STIs commonly affect the young and adolescent groups [5, 6]. The economic fate of a country depends on this segment of the population [7]. According to the World Health Organization report, about 374 million new cases of STIs have occurred every year [8]. The Sab Saharan African countries have the highest STI cases, which contribute about 93 million cases every year [9]. In developing countries, there is no adequate access to equipment, laboratory services, or skilled professionals to identify the etiologic cause of STIs, which increases the burden of STIs [10]. The prevalence of delayed treatment for STIs in sub-Saharan Africa is varying across the countries, with 23.1% in Durban, South Africa [11], 42% in Laos [12], 64% in Ghana [13], 58% in Uganda [14], and 67% in Ethiopia [15].

The impact of untreated STIs (both curable and incurable STIs) in developing countries accounts for about 17% of all economic losses, and it also causes medical complications such as cervical inflammation, genito-urinary infection, disability among infants, ectopic pregnancy, infertility, cardiovascular disorders, and increases the risk of human immunodeficiency virus transmission [16–19]. Early detection and management of STIs can be the best solution to stopping the chain of transmission and complications [20]. To overcome the impact of STIs, the World Health Organization (WHO) set a strategy on STIs in 2016 to end the epidemic of both curable and incurable STIs between 2016 and 2021 [21].

Having the symptoms promotes patients to seek treatment for STIs, which is the major reason for visiting health institutions [22]. But some symptoms cause people to embrace seeking treatment, which is the reason why a delay in the treatment of STIs [23]. Self-medicating behavior, physical and financial affordability of health services, poor knowledge about STIs, and low socioeconomic status are the risk factors or gaps for the timing of treatment seeking (delayed treatment) for both curable and incurable STIs [24–27]. As previous studies reported, there is an inconsistence of prevalence and determinants of delayed treatment for STIs across the country of Saharan Africa, and there is no comprehensive evidence for decision-makers. Even though detection and treatment of STIs a very important to reduce the burden of STIs,

nothing is known about the pooled prevalence and determinants of delayed treatment for STIs in sub-Saharan Africa for evidence-based interventions, preventive strategies, and policy making. Therefore, the current study aimed to assess the pooled prevalence and determinants of delayed treatment for STIs in sub-Saharan Africa using a systematic review and meta-analysis.

## Methods

### Searching strategy

We followed the Preferred Reporting Items for Systematic Review and Meta-Analysis (PRISMA) guidelines [28]. To do this systematic review and meta-analysis, an extensive search of the literature was done from electronic search engines like Medline via PubMed, HINARI, Embase, Scopus, Cochrane Library, Science Direct, and from websites like Google Scholar based on predefined inclusion criteria and entry terms from March 7, 2023, to March 21, 2023. The time coverage of published and unpublished literature included for this study was articles published or studied at any time period (starting from the very beginning until the last searching date). We used keywords and Medical Subject Heading (MeSH) terms or synonyms for sexually transmitted diseases together with [Title/Abstract] and combined these with Boolean operators for searching using "Delay AND Health Care Seeking OR Treatment Seeking AND Sexually Transmitted Diseases OR STDs OR Sexually Transmitted Infections OR STIs OR Venereal Diseases."

### Eligibility criteria

**Inclusion criteria.**

- Those articles that addressed either the proportion of delay in health care seeking or determinants of delay in health care seeking for STIs were included.

- Articles conducted by cross-sectional studies, prospective or retrospective cohort studies, experimental studies, and case-control studies were potentially illegible for this systematic review and meta-analysis.

- Articles written in English and done in sub-Saharan African countries were included.

- All sexually active populations (both male and female) were included.

**Exclusion criteria.** • Articles with no outcome of interest or full text were excluded.

- Short reports, case series, case reports, letters to the editor, comments, and opinions were excluded from the Meta analysis.

### Screening and study selection

After searching with predefined entry terms, we exported studies from each electronic search engine to EndNote-8. Then the duplicated studies were removed, and study screening was done by title, then by abstract, and finally by full text.

### Outcome of the study

The outcome of interest for this study was a delay in health care seeking for sexually transmitted disease (STD). The majority of the articles were defined as "delayed treatment for STIs" when patients visited health facilities after 7 days of the recognition of symptoms and otherwise early treatment of STIs.

## Data extraction

Data were extracted for each included study, like the name of the first author, year of study, recruitment site, outcome definition, types of study participants, region, study design, sample size, delayed treatment, quality assessment, proportion of delayed treatment for STI, determinants with OR, and their 95% CIs, by using the data extraction format. Data extraction from the selected articles was done independently by five reviews.

## Quality assessment

The quality of the studies was assessed using a standardized tool adapted from the Joana Brigg's Institute critical appraisal checklist for cross-sectional studies [29, 30]. Articles with a total quality score of more than 50% were labelled as paper-qualified articles with a low risk of bias. During the quality appraisal, these discrepancies were solved by discussion.

*Data synthesis and analysis.* Both systematic reviews and meta-analyses were done using STATA 17 software. In the qualitative part (systematic review), all of the eligible articles were summarized according to publication year, country, and study. A meta-analysis was conducted to determine the overall pooled prevalence of delay in healthcare seeking for sexually transmitted diseases. Publication bias was assessed by a funnel plot and Egger's regression tests. The odds ratio of 95% CL was used to identify the factors contributing to delayed treatment for STI in Sub-Saharan Africa. The prevalence of delayed treatment for STIs and other factors was reported by both the text and the forest plot. To estimate the pooled prevalence, we calculate the proportion of delayed treatment for STI using outcome and sample size. Then, using the metaprop STATA command, the pooled prevalence was estimated. We consider the exclusion of articles to minimize the effect of a very small proportion in the pooled prevalence. To conduct the pooled odds ratio, first we calculate the log odds of the odds ratio (Log OR) and the standard error of the OR (ln of the upper confidence interval minus ln of the lower confidence interval) using Excel. Then, using the "metan" STATA command, the pooled effect size (OR) was estimated. Heterogeneity was assessed both graphically (Galbraith plot and Forest plot) and statistically using $I^2$ statistics. A random-effects model, exclusion of articles, subgroup analysis by region, recruitment site of the participants, and study period were also conducted. Sensitivity analysis was also conducted to evaluate the effect of each study on the pooled prevalence. The magnitude of statistical heterogeneity between studies was assessed using $I^2$ statistics, and values of 25, 50, and 75% were considered to represent low, medium, and high, respectively.

## Ethical declaration

We used the published articles for further analysis. Our participants were the articles. Because of this, consent was not obtained from the study participants.

## Results

### Identification of the studies

In the current systematic review and meta-analysis, a total of 1989 articles were identified in the databases, websites, and registers. Of the total searched articles, 555 and 670 were duplicated and unrelated, respectively. Furthermore, 216 articles were screened for selection. Finally, after a detailed and very serious evaluation of the articles, 13 articles were included for qualitative and quantitative synthesis, as summarized in the PRISMA-2020 flow diagram (Fig 1).

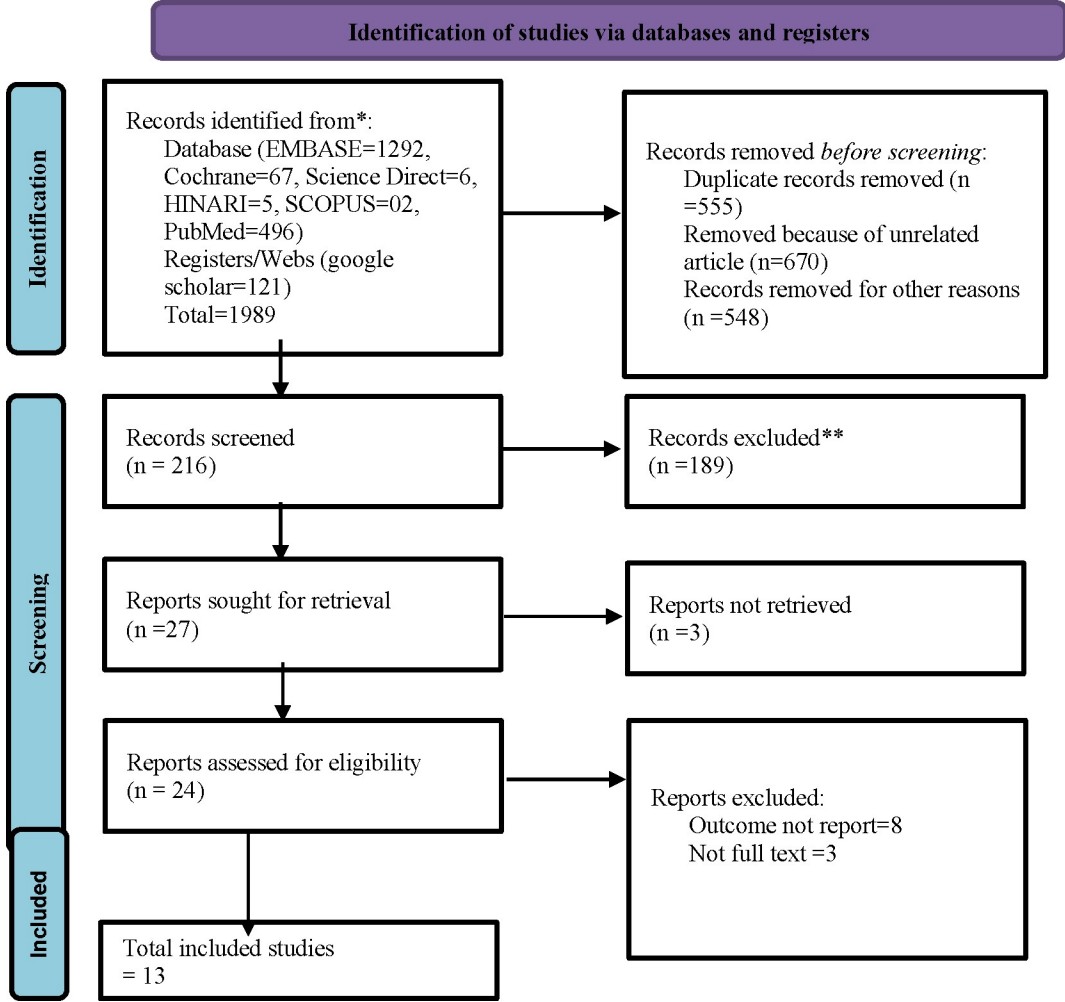

**Fig 1. PRISMA flow diagram of study selection for pooled prevalence of delay treatment for STI and its factors in Sub-Saharan Africa.**

**Characteristics of the included studies.** n the total of 13 included studies, 4 (30.8%) were published within a five-year period. Of those latest published articles, 75% were in the eastern parts of Africa. All included studies were conducted using a cross-sectional study design. In all articles, 46,722 participants participated, and the minimum and maximum sample were 58 [31] and 31,079 [32], respectively. Furthermore, the majority of studies (84.6%) define delayed treatment as a delay when patients seek treatment after 7 days following a symptom (**Table 1**).

**The pooled prevalence of delayed treatment for STI in sub-Saharan Africa.** As a random effect model showed, the pooled prevalence of delayed treatment for STI in sub-Saharan African countries was 47% (95% CI: 42%–51%). The heterogeneity among the studies was statistically significant ($I^2$ = 98.34, p-value < 0.001) (**Fig 2**). As Egger's statistical test and funnel plot showed, there was no publication bias in the included studies ($\beta$ = 0.56, P-value = 0.084) (**Fig 3**).

**Sensitivity analysis.** As the random effects model of sensitivity analysis evidenced, no studies excessively influence the overall pooled prevalence of delayed treatment for STIs (**S1 File**).

**Table 1. Characteristics of the included studies for delayed treatment of STI in sub-Saharan Africa.**

| S. No | Author | Study period | Types of participant | Recruitment sites | Delayed STI Rx | region | Quality | Sample size | Prevalence |
|---|---|---|---|---|---|---|---|---|---|
| 1 | A. Meyer-Weitz et al [33]. | 1997 | all population groups | treatment center | after 7 days following symptom | Southern Africa | low risk | 1482 | 41% |
| 2 | Meyer-Weitz Anna etal [34]. | 2000 | Adolescents | treatment center | after 7 days following symptom | Southern Africa | low risk | 292 | 44% |
| 3 | Matovu et al [35]. | 2008 | FSWs & T/drivers | community based | after 2 days of symptom | East Africa | low risk | 520 | 23% |
| 4 | Tsadik et al [36]. | 2017 | All population groups | treatment center | after 7 days following symptom | East Africa | low risk | 424 | 57% |
| 5 | Leichliter et al [37]. | 2005–2006 | Among men | treatment center | after 7 days following symptom | Southern Africa | low risk | 601 | 44% |
| 6 | Misganew. D et al [38]. | 2022 | all population groups | treatment center | after 7 days following symptom | East Africa | low risk | 404 | 60% |
| 7 | Moris. Etal [39]. | 2006 | Truck drivers | community based | after 7 days following symptom | East Africa | low risk | 58 | 22% |
| 8 | M Nyalela. et al [40]. | 2015 | All population groups | treatment center | after 7 days following symptom | Southern Africa | low risk | 134 | 23% |
| 9 | Lewis JJ. Etal [41] | 1998–2000 | All population groups | community based | after 7 days following symptom | Southern Africa | low risk | 9480 | 50% |
| 10 | Nyalela.M [42] | 2015 | All population groups | treatment center | after 6 days following symptom | Southern Africa | low risk | 134 | 53% |
| 11 | Agambire. R, Clerk.C [43]. | 2007 | All population groups | treatment center | after 7 days following symptom | Southern Africa | low risk | 185 | 64% |
| 12 | Ogola.P[32] | 2014 | All population groups | community based | after 7 days following symptom | Eastern Africa | low risk | 31079 | 57% |
| 13 | Voeten HA etal [44]. | 1999 | All population groups | community based | after 7 days following symptom | Eastern Africa | low risk | 1929 | 61% |

**Subgroup analysis by regions.** To resolve the heterogeneity between studies, in addition to using a random effect model, a sub-group analysis by region of study or country was conducted. The subgroup analysis by region or country showed that the highest pooled prevalence of delayed treatment for STI in Sub-Saharan Africa was in the eastern part of Africa, which was 50% (95% CI: 41%–59%, $I^2$ = 98.42, p<0.001) (**Fig 4**).

**Subgroup analysis by study period.** Subgroup analysis was also carried out to manage heterogeneity using the study period, and thus the maximum pooled prevalence of delayed treatment for STI in sub-Saharan Africa was studies conducted after 2010, which was 50% (95% CI: 42%–59%, $I^2$ = 98.42, p<0.001) (**Fig 5**).

**Subgroup analysis by recruitment sites.** Another subgroup analysis was also conducted on delayed treatment for STIs by recruitment sites. Articles were reported about the recruitment site in terms of the treatment center and the community. Participants in the community-based study were interviewed about their previous treatment-seeking behavior, such as delay and early. Thus, the result showed that the pooled prevalence of delayed treatment for STI was higher in the treatment center (48% 95% CI, 41%–56%, $I^2$ = 98.42, p<0.001) (**Fig 6**).

**Factors associated with delayed treatment for STI in sub-Saharan Africa.** As shown in **Table 2**, the pooled analysis of factors revealed that rural residents having poor knowledge about STIs, perceived as STIs not serious, were statistically significant variables for delayed treatment for STIs in Sub-Saharan Africa.

The random effect model estimate showed that the odds of delayed treatment for STI among those who lived in rural areas were 1.44 times higher than those among urban residents (OR = 1.44, 95% CI: 1.03–2.01), with $I^2$ = 39.3% and a p = 0.19 (**Table 2, S2 File**). Having poor

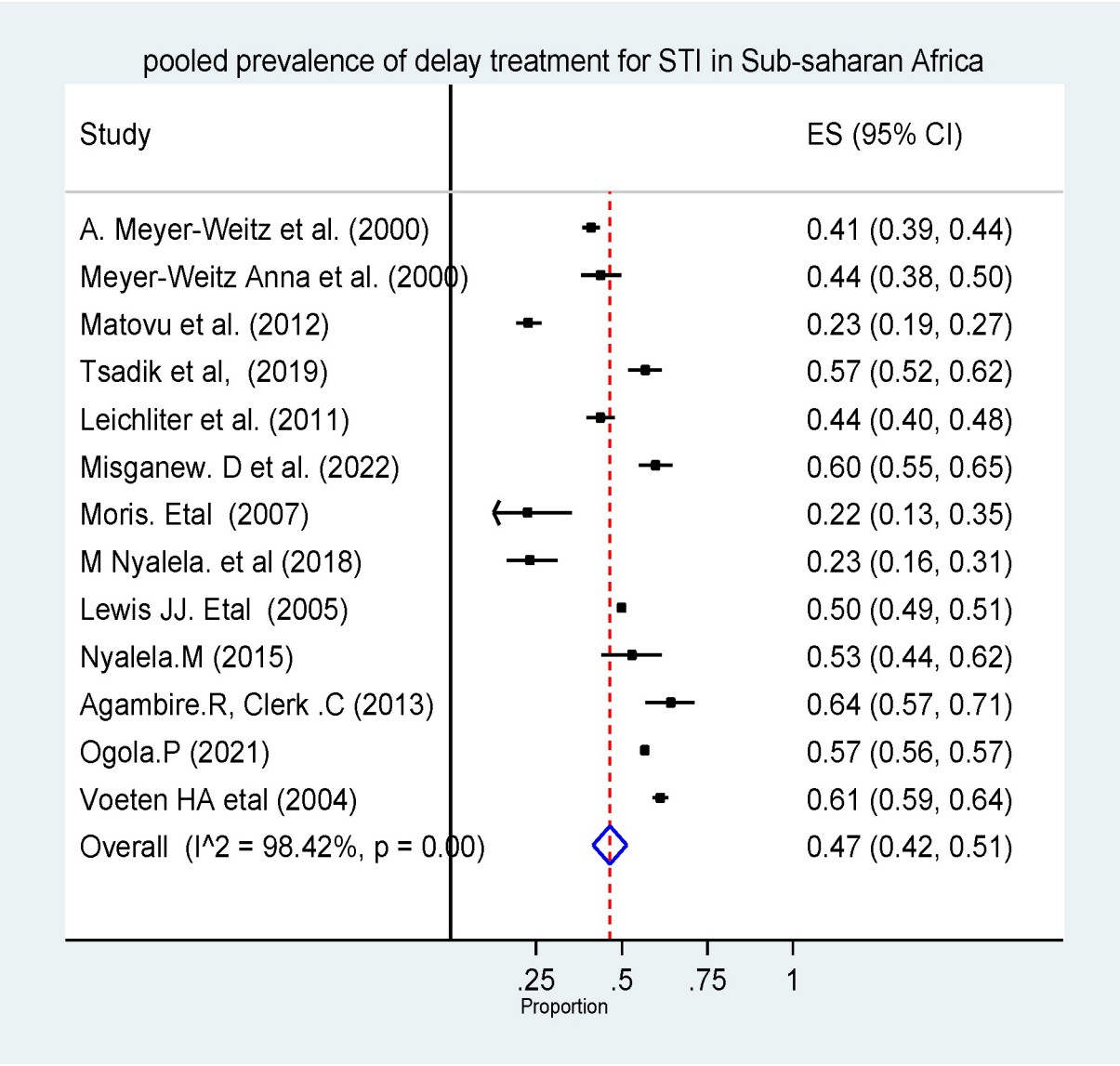

**Fig 2. The pooled prevalence of delayed treatment for STI in sub-Saharan African countries.**

knowledge about STI was also 1.49 times more likely to delay treatment for STI than having good knowledge (OR = 1.49, 95% CI: 1.04–2.13), with $I^2$ = 93.1%, p-value <0.001 (**Table 2, S3 File**). Furthermore, the odds of delayed treatment for STIs among participants who perceive STIs as not serious were 2.1 times more likely than those perceived as STIs serious (OR = 2.1, 95% CI: 1.86–2.36), with $I^2$ = 73.7%, p-value = 0.022 (**Table 2, S4 File**). In the systematic review or qualitative synthesis, the odds of delayed treatment for STI among participants who had no education, primary education, or secondary education were 3.7 times (OR = 3.17, 95% CI: 2.23–4.2), 4.1 times (OR = 4.1, 95% CI: 3.4–5.1), and 1.57 times (OR = 1.57, 95% CI: 1.1–2.76) higher than those above secondary education, respectively [38]. Furthermore, participants who had misconceptions about STD causes were 1.39 (OR = 1.39, 95% CI: 1.12–1.72) times more likely to delay treatment for STIs than those who had no misconceptions about STD causes [33].

```
Number of studies =  13                              Root MSE    =  7.199

        Std_Eff |    Coef.    Std. Err.      t     P>|t|    [95% Conf. Interval]
     -----------+----------------------------------------------------------------
          slope |  .569404    .021853     26.06    0.000     .5213059    .6175021
           bias | -5.043305   2.65072     -1.90    0.084     -10.8775    .7908915

Test of H0: no small-study effects              P = 0.084
```

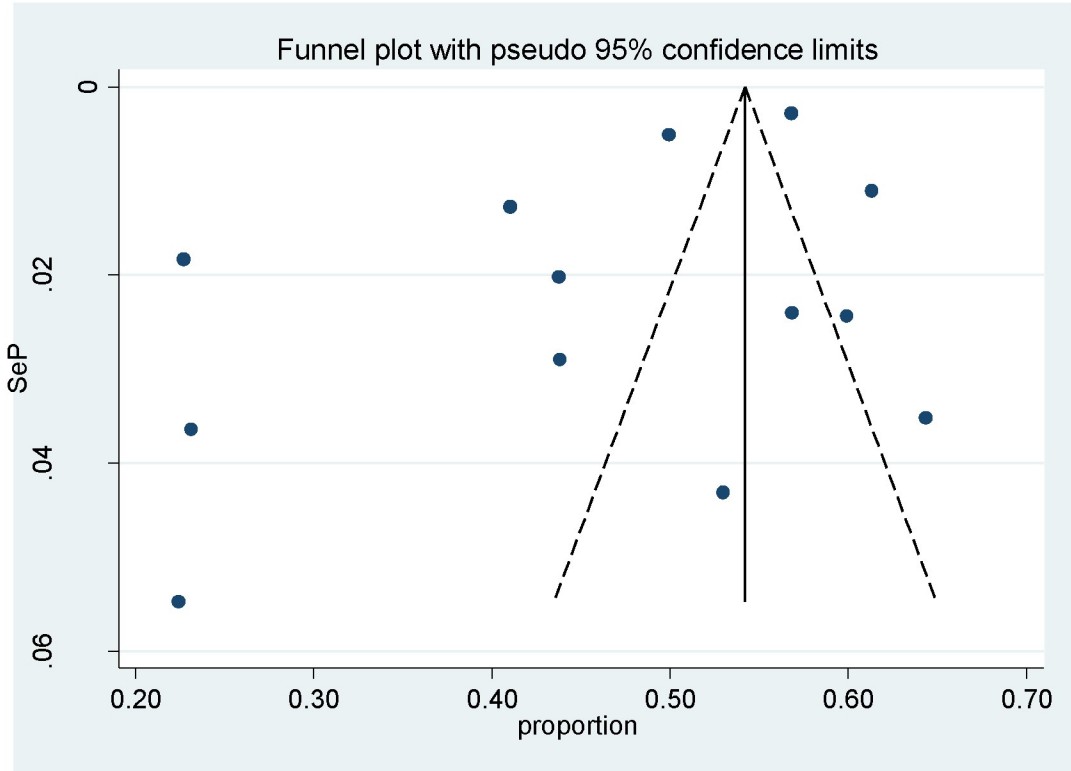

**Fig 3. Funnel plot to assess the publication bias for the study of delayed treatment for STI in sub-Saharan African countries.**

## Discussion

In this study, an attempt has been made to assess the pooled prevalence of delayed treatment for STI and its determinants in Sub-Saharan African countries. Despite considerable heterogeneity ($I^2$ = 98.42%, P = <0.001), the pooled prevalence of delayed treatment for STIs was higher among studies conducted after 2010 than before 2010, which was 50% (95% CI: 42%–59%). This may be because the COVID-19 pandemic negatively affects the screening practice for hypertension. Despite considerable heterogeneity ($I^2$ = 98.42%, P = <0.001), this study also revealed that the pooled prevalence of delayed treatment for STI in Sub-Saharan Africa was 47% (95% CI: 42%–51%), which was higher than a study conducted in Singapore (27%) [45], in New York City 30% [46], in China (32% women and female 25%) [47], in five urban STD clinics in the United States (35% of men and 37% of women) [48]. The possible reason for this discrepancy might be the difference in socioeconomic condition, health service access, culture, and awareness between American and Asian countries and Sub-Saharan African countries. In

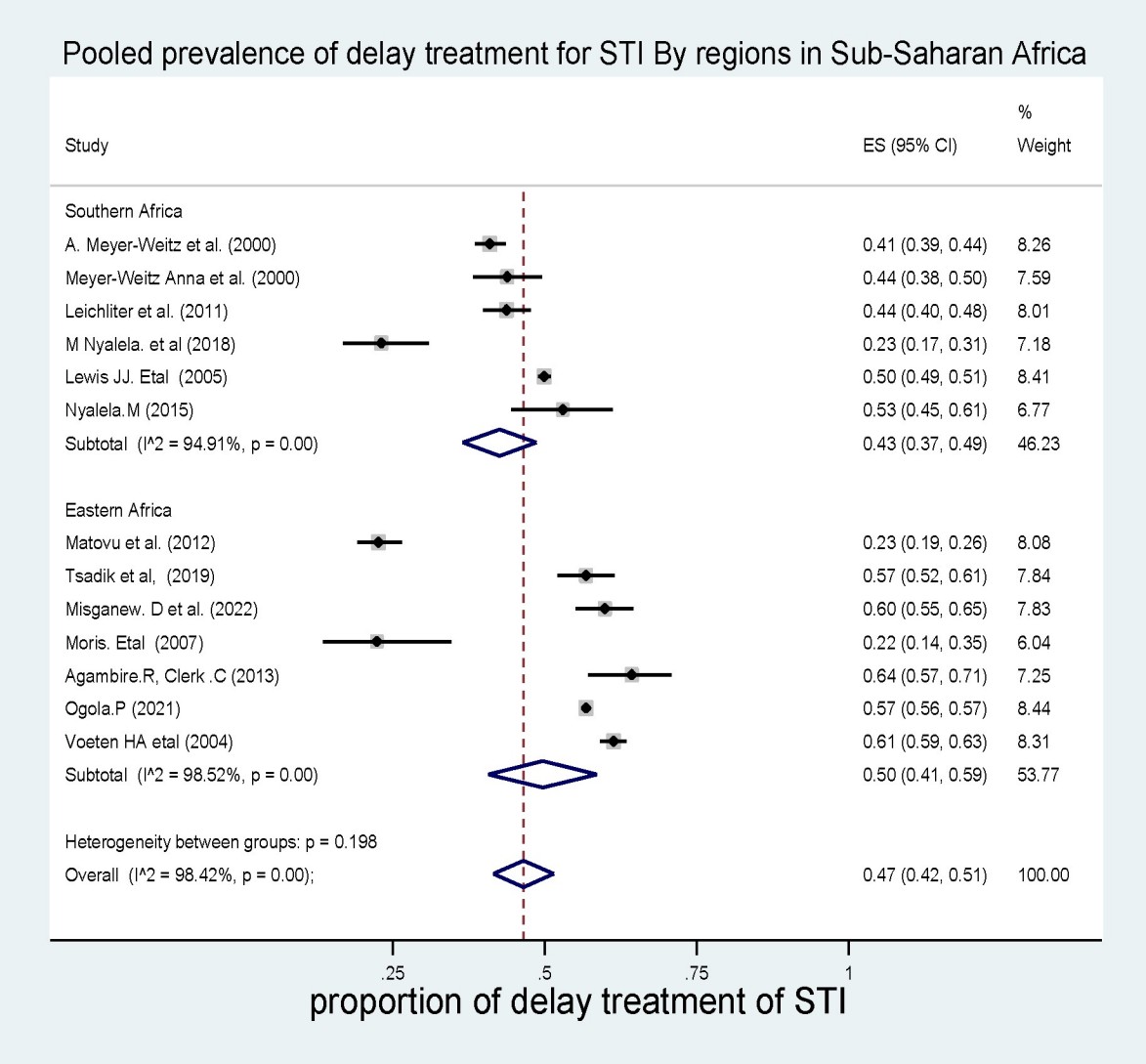

**Fig 4. Pooled prevalence of delayed treatment for STI in sub-Saharan African countries by regions.**

contrast, the pooled prevalence of delayed treatment for STI in sub-Saharan African countries was lower than a study conducted in Vietnam [49]. The possible reason may be the difference in sociocultural and study periods. The second reason might be because of the variation in the study population. In the study of Vietnam, the study participants were only women. Women had fewer severe symptoms of STIs than men, and because of this, they may delay getting treatment and overestimate the finding [50].

Regarding factors associated with delayed treatment for STIs, perceiving STI as not serious was associated with higher odds of delayed treatment for STI than perceiving STI as serious, with heterogeneity between studies ($I^2$ = 73.7%, p = 0.022). This was supported by a study conducted among adolescent African-American females [51]. This might be due to people perceiving STIs as not serious being more vulnerable to refraining from STI treatment for a persistent period of time or negatively affecting their decision to seek health care than those who perceive STIs as serious.

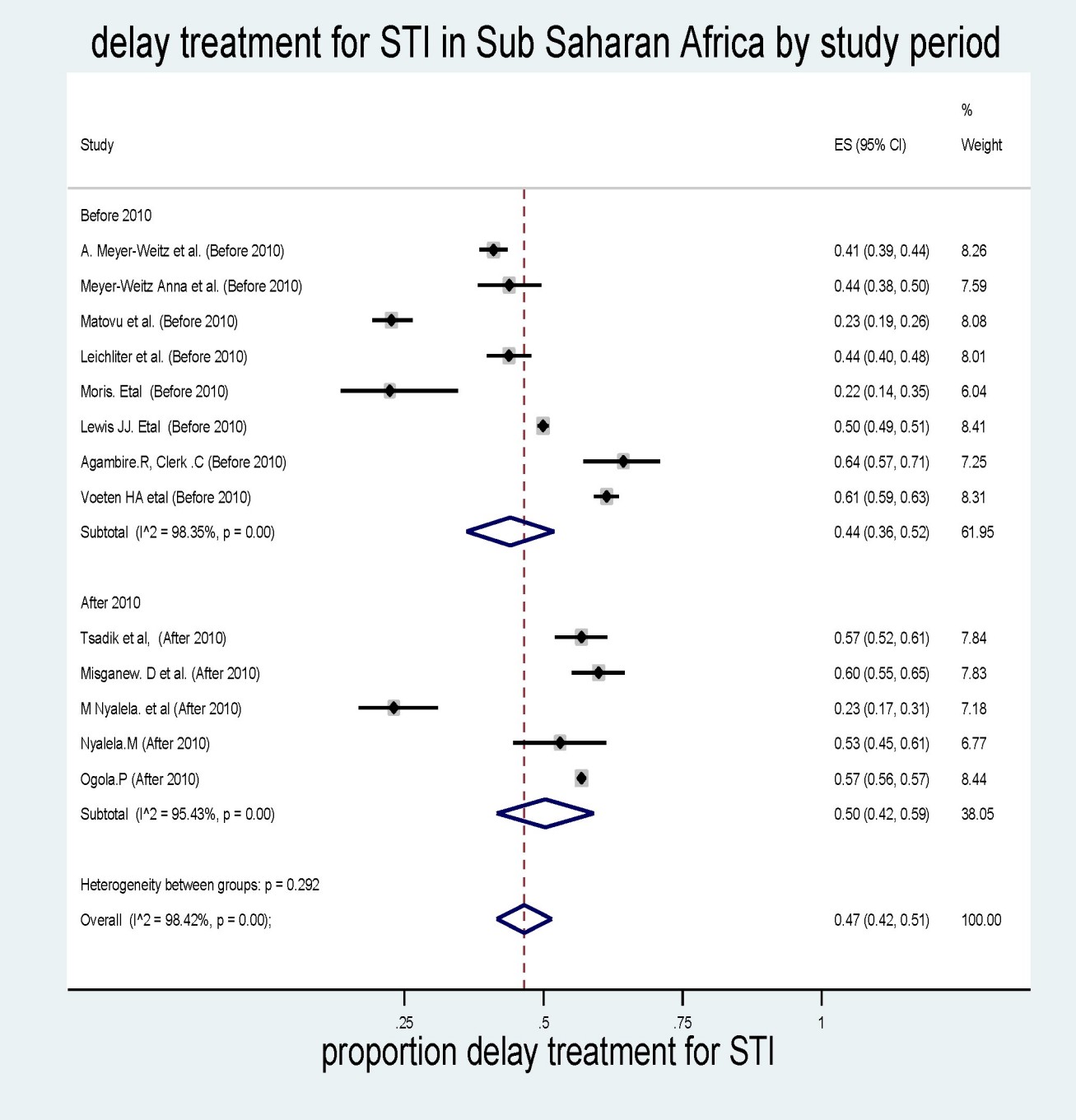

**Fig 5. Pooled prevalence of delayed treatment for STI in sub-Saharan African countries by study period.**

Despite considerable heterogeneity between the studies ($I^2 = 93.1\%$, p<0.00l), poor knowledge about STIs was also significantly affecting treatment-seeking behavior. This finding was consistent with studies conducted in Singapore [45] and India new Delhi [52]. The possible reason might be associated with poor knowledge about STI complications, signs and symptoms, the importance of early treatment, and curability, which may hinder the early treatment-

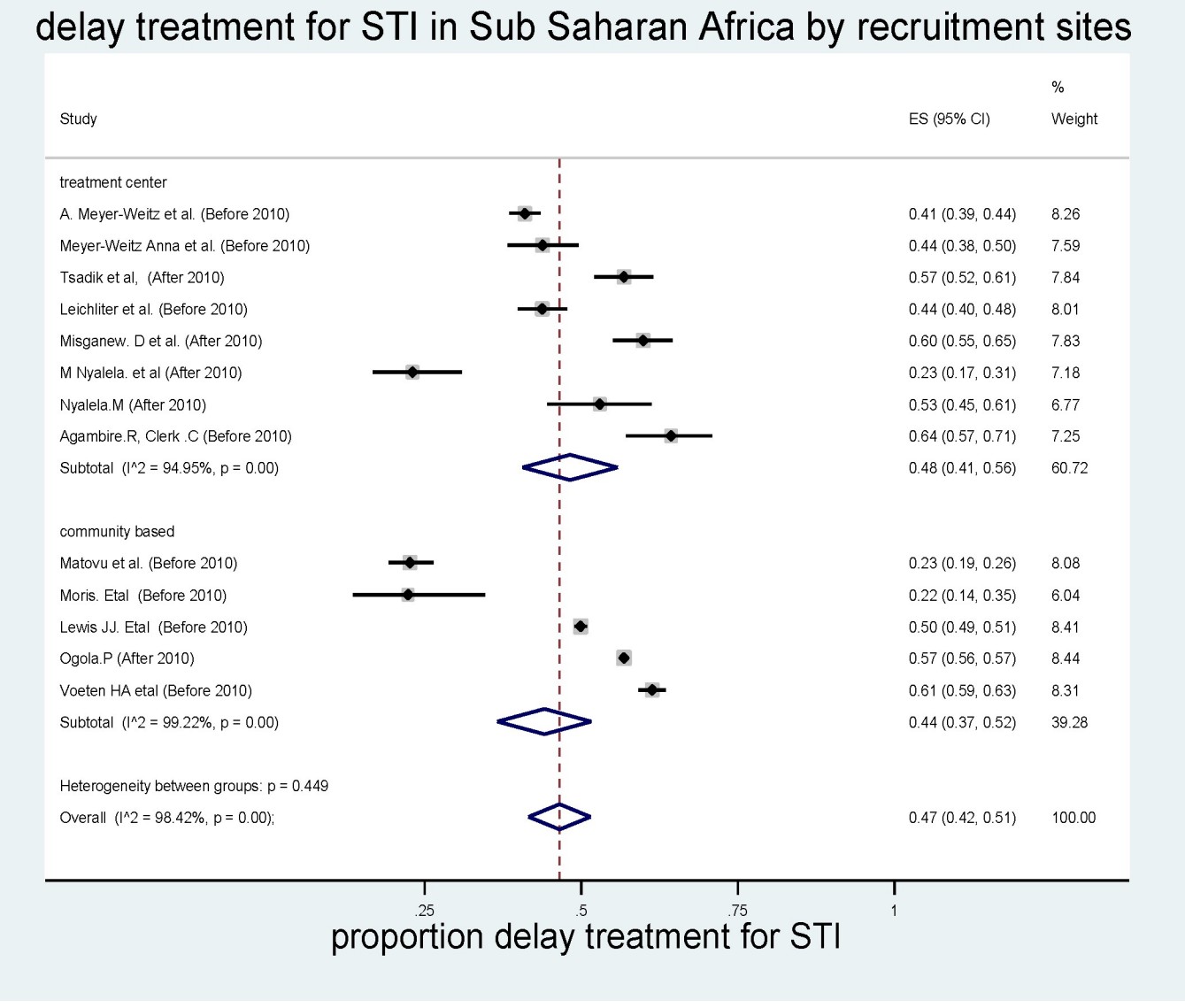

**Fig 6. Sub group analysis of delayed treatment for STI in sub Saharan Africa by recruitment site.**

seeking behavior of STIs. It is also well known that knowledge is an important factor in early recognizing the disease and taking steps to treat or prevent it, but poor knowledge could delay the treatment [52].

Additionally, participants from rural residences were more likely to delay treatment for STIs than those who lived in urban areas, with no heterogeneity between studies ($I^2$ = 39.3%, p = 0.199). This finding was consistent with a study conducted in the Netherlands [53]. This might be due to the fact that clinics in rural areas are not more accessible than in urban ones. The other possible reason might be the difference in health information and media access and socio-economic status.

In the qualitative synthesis or systematic review, participants who had no education, primary education, or secondary education were more likely to delay the treatment of STIs (26).

**Table 2. Factors associated with delayed treatment for STI in sub Saharan Africa.**

| Variable | | OR(95%CI) | Heterogeneity (I²,P-value) | Total studies | Sample size |
|---|---|---|---|---|---|
| Sex | Female | 1.21(0.98, 1.49) | 84.3%, 0.002 | 3 | 2032 |
| | Male | 1 | 1 | | |
| Perception about STI | Not serious | 2.1(1,86, 2.36)* | 73.7%, 0.022 | 3 | 1091 |
| | Neutral | 0.76 (0.46, 1.23) | 93.7%, <0.001 | 2 | 483 |
| | Serious | 1 | 1 | | |
| Numbers of sexual partner | > = 2 | 1.42 (0.36,5.32) | 0%, 0.332 | 2 | 483 |
| | < = 1 | 1 | 1 | | |
| knowledge | Poor | 1.49 (1.04, 2.13)* | 93.1%, <0.001 | 2 | 483 |
| | Good | 1 | 1 | | 2846 |
| Fear of stigma | Yes | 2.08 (0.67, 6.47) | 0%, 0.766 | 2 | 483 |
| | No | 1 | 1 | | |
| Residence | Rural | 1.44 (1.03, 2.01)* | 39.3%, 0.199 | 2 | 313 |
| | Urban | 1 | 1 | | |
| Marital status | Married | 1.18 (0,81, 1.71) | 72.3%, 0.0571 | 2 | 483 |
| | Un married | 1 | 1 | | |
| Self-treatment | Yes | 0.6 (0.46, 1.78) | 23.8%, 0.252 | 2 | 736 |
| | No | 1 | 1 | | |

he possible reason for this finding might be that the educational level of people increased, which could improve their knowledge, attitude, and socio-economic status, which leads to better health-seeking behavior.

Misconceptions about the cause of STDs were also a risk factor for delayed STI treatment (38). This might be treatment-seeking behavior among those who misunderstand STIs, which causes an intention to treat themselves at home using different modalities instead of seeking modern medicine at health facilities [50]. The strength of the study was that the data report adhered to guidelines outlined in the PRISMA-P 2020 statement protocol. Even though several attempts have been made to overcome the heterogeneity between the studies, it has not been managed. This was the limitation of this study. But this study has several implications, such as providing comprehensive evidence for designing policies and strategies. It is also used to promote reproductive health in sub-Saharan African countries.

## Conclusion

The pooled prevalence of delayed treatment for STIs in Sub-Saharan African countries was high. Poor knowledge, attitude, and educational status affect the treatment delay for STIs. Thus, improving knowledge, educational status, and attitude are highly recommended.

## Supporting information

**S1 Checklist.**
(DOCX)

**S1 Data.**
(XLSX)

**S1 File. Sensitivity analysis for the study of delayed treatment for STI in sub-Saharan African countries.**
(DOCX)

**S2 File. Effect of rural residence on delay treatment for STI in sub-Saharan African countries.**
(DOCX)

**S3 File. Effect of poor knowledge on delayed treatment for STI in sub-Saharan African countries.**
(DOCX)

**S4 File. Effect of perceived STI not serious on delayed treatment for STI in sub Saharan African countries.**
(DOCX)

## Acknowledgments

The authors acknowledged university of Gondar for providing systematic review and meta-analysis training.

## Author Contributions

**Conceptualization:** Muluken Chanie Agimas, Milkias Solomon, Dessie Abebaw Angaw.

**Data curation:** Muluken Chanie Agimas.

**Formal analysis:** Muluken Chanie Agimas, Milkias Solomon, Nebiyu Mekonnen Derseh.

**Investigation:** Muluken Chanie Agimas.

**Methodology:** Muluken Chanie Agimas, Daniel Alayu Shewaye, Dessie Abebaw Angaw, Nebiyu Mekonnen Derseh.

**Software:** Muluken Chanie Agimas, Nebiyu Mekonnen Derseh.

**Supervision:** Muluken Chanie Agimas, Dessie Abebaw Angaw.

**Validation:** Muluken Chanie Agimas, Nebiyu Mekonnen Derseh.

**Visualization:** Muluken Chanie Agimas, Nebiyu Mekonnen Derseh.

**Writing – original draft:** Muluken Chanie Agimas, Nebiyu Mekonnen Derseh.

**Writing – review & editing:** Muluken Chanie Agimas.

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
