## [Decision Letter · Decision Letter 0]

4 Jan 2024

PONE-D-23-13983Prevalence of delayed treatment for sexually transmitted infections and its determinants in Sub-Saharan Africa. A systematic review and Meta-analysisPLOS ONE

Dear Dr. chanie,

Thank you for submitting your manuscript to PLOS ONE. After careful consideration, we feel that it has merit but does not fully meet PLOS ONE’s publication criteria as it currently stands. Therefore, we invite you to submit a revised version of the manuscript that addresses the points raised during the review process.

 Please respond to each and every comment of the 2 reviewers, with either accommodation or a rebuttal.

We look forward to receiving your revised manuscript, and hope to see your MS in published form.

Kind regards,

D. William Cameron, MD

Academic Editor

PLOS ONE

Journal Requirements:

5. Please include your tables as part of your main manuscript and remove the individual files. Please note that supplementary tables (should remain/ be uploaded) as separate "supporting information" files.

Reviewers' comments:

Reviewer's Responses to Questions

**Comments to the Author**

1. Is the manuscript technically sound, and do the data support the conclusions?

Reviewer #1: Yes

Reviewer #2: Partly

2. Has the statistical analysis been performed appropriately and rigorously? 

Reviewer #1: Yes

Reviewer #2: Yes

3. Have the authors made all data underlying the findings in their manuscript fully available?

Reviewer #1: Yes

Reviewer #2: Yes

4. Is the manuscript presented in an intelligible fashion and written in standard English?

Reviewer #1: No

Reviewer #2: Yes

5. Review Comments to the Author

Reviewer #1: Systematic review

ABSTRACT:

-ABSTRACT, result: a bit confused as no reference group was mentioned (from result in main text it’s above secondary school). Also, the terms for “no education”, “primary education”, “secondary education” should be revised. Methods of ABSTRACT did not mention how the OR was calculated.

-ABSTRACT, conclusion: risk factors duplicated with results. Suggest to summarize or highlight or just delete it

INTRDOUCTION:

-The authors are highly recommended to trim the introduction, especially the overlapping parts or sentences. For instance, lines 77-80 about the STI burden could be shortened by selecting either reference source instead of listing both.

- HIV may not be included as STI as the symptom presentation and reasons for delayed treatment was quite different from common STI. And please double check if HIV was excluded from the literature search. “Bacteria, viruses, and parasites are responsible for STIs such as gonorrhea, chlamydial infection, syphilis, trichomoniasis, chancroid, genital herpes, human immunodeficiency virus infection, and hepatitis B infection.”

-“Among all STIs, syphilis, gonorrhea, chlamydia, and trichomoniasis are curable (8)” They are not the only ones, but the bacterial STI curable and with higher disease burden. So this sentence is misleading with “all” included.

-“Thus, the transmission rate of the human immunodeficiency virus … ” don’t understand this sentence. Do the authors mean STI was a risk factor for HIV infection?

METHODS:

-instead of the time period for performing systematic review, it’s more important for readers to know the time coverage of literature published. Did the literature search started from very beginning (say 1980) up till March 2023?

-please suggest if the eligibility criteria included study design. For calculation of pooled delayed treatment proportion, should RCT be excluded?

-outcome of the study: the ‘time taken” for defining delayed treatment was not clear. So how long of interval was defined as “yes”? actually, I believe the definition of delayed treatment varied across studies which the authors could not control. If that’s true, please add a paragraph in RESULTS to summarize the definition of delayed treatment for STI used in studies. This would be useful reference for future studies.

-has the systematic review registered? Please mention this. If no, please contact editor on how to handle this.

RESULTS:

-‘handling heterogeneity’: please cite the table/figure for this. Also, what’re the results for “Sensitivity analysis, and subgroup analysis were performed to assess the heterogeneity” ? Without results, the sentence should belong to METHODS.

-subgroup analysis by year should use study year instead of publication year. A study could be conducted before 2010 but the publication year was after 2010.

-Table 2 should add columns of list of STI included in study, the definition of delayed treatment, recruitment site (e.g. STI treatment clinics, NGOs, online platform) and/or participants (e.g. heterosexual male, heterosexual female, MSM, FSW, clients of FSW, PWID).

-if possible, authors are recommended to perform subgroup analysis by type of STI, and type of recruitment sites.

DISCUSSION:

-The interpretation of subgroup analysis by year may need caution with some notes. Comparison of studies by year should be, at least, based on the same list of STI. So that we could understand if there was actual increase of delayed treatment over time. Please revise this sentence: “Delayed treatment for STIs was higher after 2010 than before 2010. This means the distribution of delayed treatment for STIs has increased over time in Sub-Saharan African countries.”

-What is “Nework 30%”?

-don’t understand “That means a study in Vietnam was conducted only among women and women who experienced less severe symptoms than men and thus may delay seeking treatment (53).” The “lower” proportion in the previous sentence did not match with this.

-duplication on the meaning for “Thus, the perception of people about STIs influences treatment-seeking behavior. This means participants perceiving STIs as not serious were at higher risk of delayed treatment for STIs in Sub-Saharan Africa.”

-line 262, “In the qualitative synthesis or systematic review …” should it be meta-analysis instead of systematic review?

Overall: please double check the whole manuscript carefully to make sure there’re no duplication of meaning in sentences nearby, no “strange” terms used (e.g. “end or advance into a sexually transmitted diseases”, which should be seroclearance or progression …, goals of the sustainable development goal [duplicated]). Please make sure the sentence describing STI is correct and precise in INTRODUCTION.

Ethical declaration: please check with editor on the suggested sentence for systematic review. “Not practical” is unlikely to be a suitable term.

Reviewer #2: COMMENTS

The authors sought to determine the prevalence of delayed treatment for STIs and its determinants in sub-Saharan Africa. A systematic review and meta-analysis. This is an extensive and ambitious review of an important subject. The authors have looked for published and unpublished studies in several databases of prevalent delayed treatment for STIs in sub-Saharan Africa. I could not clearly consider the expertise of the authors.

This is an important paper. There are parts in the introduction which would need considerable attention before the manuscript can be considered for publication.

1. In line 68 - 70, consider rephrasing the sentences and avoid starting the paragraph with “Especially”. The following sentence “These youth and adolescents are the productive segment of the population and thus the economic fate of a country depends on the youth and adolescent age group a population” requires rephrasing to make sense.

2. In line 72 – 73, can the authors state all and accurate reasons why STIs are common in developing countries which should include management of STIs which is mostly syndromic approaches.

3. In line 74 – 76, the argument is not correct as most of the STIs are asymptomatic, and patients don’t know that they have an infection. I encourage the authors to present factual points on why treatment of STIs is not early.

4. In line 80 – 81, the authors referenced a figure which I could not review.

5. In line 98, not all STIs can be cured, the sentence is inaccurate.

6. The introduction does not clearly present the background on this topic to show why there is gap on timing of treatment for STIs. Also, it’s not clear if authors are focusing on any STI, curable and incurable. Considerable work might need to be done to have a well written and factual introduction.

7. For statistical analysis, may the authors state how the meta-analysis was done to obtain a pooled prevalence (proportion). Were the data transformed in any way? What assumptions were made, e.g. about the distribution of values? There are some considerations, e.g. very small proportions for prevalence of delayed treatment, which can present problems with the logit transformation using inverse variance methods (e.g. Barendregt JJ, et al. J Epidemiol Commun Health 2013;67:975-78), or possible unreliability with some transformations under certain conditions (e.g. Schwarzer G, et al. Res Synth Methods 2019;10:476

8. Statistical heterogeneity – the authors need to pay much more attention to the extreme heterogeneity in how to investigate it in the methods, to reporting it in the results and to interpretation of these average values in the Discussion. The I-squared value is given in forest plots and is mostly close to 100%. However, there is no mention of this in the Discussion. Prediction intervals, based on tau-squared, might be an additional way of displaying the between study variability, although with such extreme heterogeneity, these are also difficult to interpret.

6. PLOS authors have the option to publish the peer review history of their article (what does this mean?). If published, this will include your full peer review and any attached files.

Reviewer #1: No

Reviewer #2: **Yes: **Dorothy Nyemba

---

## [Author Response · Author response to Decision Letter 0]

15 Jan 2024

dear reviewer and editor we have submit revised manuscript point by point.

---

## [Decision Letter · Decision Letter 1]

2 Feb 2024

PONE-D-23-13983R1Prevalence of delayed treatment for sexually transmitted infections and its determinants in Sub-Saharan Africa. A systematic review and Meta-analysisPLOS ONE

Dear Dr. Agimas,

Thank you for submitting your manuscript to PLOS ONE. After careful consideration, we feel that it has merit but does not fully meet PLOS ONE’s publication criteria as it currently stands. Therefore, we invite you to submit a revised version of the manuscript that addresses the points raised during the review process.

Please address the reviewer's points from the first round of comments as well as this one this time, one by one and completely.

We look forward to receiving your revised manuscript.

Kind regards,

D. William Cameron, MD

Academic Editor

PLOS ONE

Additional Editor Comments:

You do need to address their first round and this round of comments. The best way to address them is to comply.

Reviewers' comments:

Reviewer's Responses to Questions

**Comments to the Author**

1. If the authors have adequately addressed your comments raised in a previous round of review and you feel that this manuscript is now acceptable for publication, you may indicate that here to bypass the “Comments to the Author” section, enter your conflict of interest statement in the “Confidential to Editor” section, and submit your "Accept" recommendation.

Reviewer #1: All comments have been addressed

Reviewer #2: (No Response)

2. Is the manuscript technically sound, and do the data support the conclusions?

Reviewer #1: Yes

Reviewer #2: Partly

3. Has the statistical analysis been performed appropriately and rigorously? 

Reviewer #1: I Don't Know

Reviewer #2: Yes

4. Have the authors made all data underlying the findings in their manuscript fully available?

Reviewer #1: Yes

Reviewer #2: (No Response)

5. Is the manuscript presented in an intelligible fashion and written in standard English?

Reviewer #1: No

Reviewer #2: (No Response)

6. Review Comments to the Author

Reviewer #1: ABSTRACT

Please remove the newly added reference group for binary variables, e.g. “as compared to urban”. My previous comment referred to education level variable only, as that included more than 2 categories.

INRODUCTION

I disagree with ‘Because of sexual behaviour, the burden of STIs is higher in developing countries.’, which could result in misinterpretation of people living in developing countries have more sexual behaviour. Please revise or remove this and the sentence that follows.

In general, please double check the grammar of the whole manuscript after the revision. Some newly revised parts do not fit well with the original sentences.

Reviewer #2: There are a number of points which were raised in the first review and not addressed in the revised submission.

1. A number of points to address in the introduction were not done.

2. In the method section, authors were asked to include the time period of the literature search and its still not defined in under the search strategy.

3. Authors were requested to define inclusion criteria (study design, participants/population of interest, etc).

7. PLOS authors have the option to publish the peer review history of their article (what does this mean?). If published, this will include your full peer review and any attached files.

Reviewer #1: No

Reviewer #2: No

---

## [Decision Letter · Decision Letter 2]

14 Feb 2024

Prevalence of delayed treatment for sexually transmitted infections and its determinants in Sub-Saharan Africa. A systematic review and Meta-analysis

PONE-D-23-13983R2

Dear Dr. Agimas,

We’re pleased to inform you that your manuscript has been judged scientifically suitable for publication and will be formally accepted for publication once it meets all outstanding technical requirements.

Kind regards,

D. William Cameron, MD

Academic Editor

PLOS ONE

Additional Editor Comments (optional):

Reviewers' comments:

Reviewer's Responses to Questions

**Comments to the Author**

1. If the authors have adequately addressed your comments raised in a previous round of review and you feel that this manuscript is now acceptable for publication, you may indicate that here to bypass the “Comments to the Author” section, enter your conflict of interest statement in the “Confidential to Editor” section, and submit your "Accept" recommendation.

Reviewer #1: All comments have been addressed

Reviewer #2: All comments have been addressed

2. Is the manuscript technically sound, and do the data support the conclusions?

Reviewer #1: Yes

Reviewer #2: Yes

3. Has the statistical analysis been performed appropriately and rigorously? 

Reviewer #1: I Don't Know

Reviewer #2: Yes

4. Have the authors made all data underlying the findings in their manuscript fully available?

Reviewer #1: Yes

Reviewer #2: Yes

5. Is the manuscript presented in an intelligible fashion and written in standard English?

Reviewer #1: No

Reviewer #2: Yes

6. Review Comments to the Author

Reviewer #1: I have no further comments.

Reviewer #2: No further comments for my side. The authors have reviewed their work as per the previous reviews and addressed all issues that were highlighted

7. PLOS authors have the option to publish the peer review history of their article (what does this mean?). If published, this will include your full peer review and any attached files.

Reviewer #1: No

Reviewer #2: No

---

## [Editor Report · Acceptance letter]

6 Mar 2024

PONE-D-23-13983R2 

PLOS ONE

Dear Dr. Agimas, 

I'm pleased to inform you that your manuscript has been deemed suitable for publication in PLOS ONE. Congratulations! Your manuscript is now being handed over to our production team.

Kind regards, 

on behalf of

Professor D. William Cameron 

Academic Editor

PLOS ONE